

# Mulberry EIL3 confers salt and drought tolerances and modulates ethylene biosynthetic gene expression

Changying Liu[1], Jun Li[2], Panpan Zhu[1,3], Jian Yu[1], Jiamin Hou[1], Chuanhong Wang[4], Dingpei Long[1], Maode Yu[1] and Aichun Zhao[1]

[1] State Key Laboratory of Silkworm Genome Biology, Key Laboratory of Sericultural Biology and Genetic Breeding, Chongqing, China
[2] Guiyang University of Chinese Medicine, Guiyang, China
[3] Bioengineering College of Chongqing University, Chongqing, China
[4] The National Key Engineering Lab of Crop Stress Resistance Breeding, School of Life Sciences, Anhui Agricultural University, Hefei, China

## ABSTRACT

Ethylene regulates plant abiotic stress responses and tolerances, and ethylene-insensitive3 (EIN3)/EIN3-like (EIL) proteins are the key components of ethylene signal transduction. Although the functions of EIN3/EIL proteins in response to abiotic stresses have been investigated in model plants, little is known in non-model plants, including mulberry (*Morus* L.), which is an economically important perennial woody plant. We functionally characterized a gene encoding an EIN3-like protein from mulberry, designated as *MnEIL3*. A quantitative real-time PCR analysis demonstrated that the expression of *MnEIL3* could be induced in roots and shoot by salt and drought stresses. *Arabidopsis* overexpressing *MnEIL3* exhibited an enhanced tolerance to salt and drought stresses. *MnEIL3* overexpression in *Arabidopsis* significantly upregulated the transcript abundances of ethylene biosynthetic genes. Furthermore, *MnEIL3* enhanced the activities of the *MnACO1* and *MnACS1* promoters, which respond to salt and drought stresses. Thus, MnEIL3 may play important roles in tolerance to abiotic stresses and the expression of ethylene biosynthetic genes.

# INTRODUCTION

Ethylene is gaseous hormone that regulates many physiological processes, including seed germination, seedling growth, leaf expansion, flower opening, senescence, and fruit development. Ethylene is synthesized from methionine by a catalysis mediated by S-adenosyl-L-methionine synthetase, 1-aminocyclopropane-1-carboxylic acid synthase (ACS), and 1-aminocyclopropane-1-carboxylic acid oxidase (ACO) (*Kende, 1993*; *Yang & Hoffman, 1984*). ACS and ACO are the rate-limiting enzymes that regulate ethylene biosynthesis, and several regulators influence ethylene production by changing the activities and gene expression levels of ACS and ACO.

The ethylene perception and signal transduction pathway have been well studied in model plants, like *Arabidopsis thaliana*, tomato (*Solanum lycopersicum*), and tobacco

Corresponding author
Aichun Zhao,
zhaoaichun@hotmail.com

# PeerJ

(*Nicotiana tabacum*). In the presence of ethylene, the endoplasmic reticulum-localized receptors (ethylene receptors, ETRs) perceive ethylene, resulting in the inactivation of constitutive triple response 1, which is the negative regulatory factor of ethylene responses (*Clark et al., 1998*; *Hua & Meyerowitz, 1998*; *Rodríguez et al., 1999*). Then, ethylene insensitive 2 (EIN2), an essential positive regulator of ethylene signaling, is dephosphorylated, and its carboxyl terminus is cleaved and enters into the nucleus, where it binds to ethylene insensitive 3/EIN3-like proteins (EIN3/EILs) (*Chao et al., 1997*; *Qiao et al., 2012*). Eventually, the activated EIN3/EILs regulate the transcription of ethylene-responsive factors (ERFs) and other downstream genes (*Alonso & Stepanova, 2004*; *Wang, Li & Ecker, 2002*).

EIN3/EILs are the key elements that initiate the ethylene-mediated downstream transcriptional cascade (*An et al., 2018*). The mutation of *EIN3/EILs* genes, *AtEIN3* (AT3G20770) and *AtEIL1* (AT2G27050), result in ethylene-insensitive performance, and plants overexpressing *AtEIN3* and *AtEIL1* show enhanced ethylene production and triple responses in *Arabidopsis*. The *ein3-1 eil1-1* double mutant completely abolishes the ethylene response in etiolated *Arabidopsis* seedlings (*Alonso et al., 2003*; *Chao et al., 1997*). The stabilities of EIN3/EILs are regulated by EIN3-binding F-box proteins (EBF1 and EBF2) in the EBF1- and EBF2-mediated ubiquitin-proteasome degradation pathway, and mutations of *EBF1* and *EBF2* accumulate EIN3/EIL proteins and display constitutive ethylene responses (*Potuschak et al., 2003*). Ethylene quickly stabilizes EIN3/EIL1 by promoting EBF1 and EBF2 proteasomal degradation, which contributes to the ethylene responses (*An et al., 2010*). In addition, the MKK9-MPK3/MPK6 cascades promote EIN3-mediated transcription in ethylene signaling by regulating the phosphorylation and protein stability of EIN3 (*Yoo et al., 2008*).

The EIN3/EILs family are plant-specific transcription factors (TFs) and bind to primary ethylene response elements (PEREs) and EIL conserved binding sequences (ECBSs) in the promoters of downstream genes involved in the response to ethylene (*Yin et al., 2010*). Thus, EIN3/EILs regulate many physiological processes, including apical hook formation, hormone responses, fruit development, abiotic stress responses, seedling photomorphogenesis, and light perception, by activating the expression of a wide range of downstream genes (*An et al., 2012*; *He et al., 2011*; *Peng et al., 2014*; *Shan et al., 2012*; *Shi et al., 2012*; *Shi et al., 2018*; *Zhu et al., 2011*). Recently, studies have focused on the functions of EIN3/EILs in abiotic stress tolerances. *Peng et al. (2014)* demonstrated that EIN3/EIL1 are essential for the enhanced ethylene-induced salt tolerance in *Arabidopsis*, and salt stress stabilizes EIN3/EIL1 proteins by promoting EBF1/EBF2 proteasomal degradation in an EIN2 independent manner. In addition, a large number of EIN3/EIL1-regulated genes that participate in salt stress responses have been identified using whole-genome transcriptome analyses, including many genes encoding reactive oxygen species scavengers. An AP2 domain-containing gene, *ESE1*, is an ethylene-modulated gene downstream of EIN3/EIL1 in the salt response (*Zhang et al., 2011*). Mutations of *EIN3* increase the sensitivity in response to water stress stimulated by polyethylene glycol (PEG) 6,000 (*Cui et al., 2015*). Genetic and biochemical analyses revealed that EIN3 proteins act as negative factors against freezing stress by repressing

the expression of C-repeat binding factors and type-A *Arabidopsis* response regulator (*ARR*) 5, *ARR7*, and *ARR15* (*Shi et al., 2012*). The functions of the EIN3/EIL1 proteins in response to heavy metal stresses have also been studied. *Kong et al. (2018)* found that cadmium (Cd) inhibits EIN3 protein degradation in *Arabidopsis*, and the *ein3-1 eil1-1* double mutant plants display an increased tolerance to Cd. EIN3 enhances root growth inhibition under Cd stress by regulating the expression of the xyloglucan endotransglucosylase/hydrolase 33 and response to low sulfur 1 genes, which are involved in cell wall modification and sulfur metabolic processes, respectively (*Kong et al., 2018*).

Mulberry (*Morus* L.) is an economically important perennial woody plant belonging to *Moraceae* of *Rosales*, which have multiple uses in silkworm rearing, ecology, pharmaceuticals, and traditional Chinese medicines (*He et al., 2013*). Mulberry adapts well to drought, salinity, waterlogging, and other abiotic stress conditions, but little is known regarding the molecular mechanisms of the tolerance. In our previous studies, the elements involved in mulberry ethylene biosynthesis and signal transduction were identified and its functions in fruit development were clarified (*Liu et al., 2014*, *2015*). However, the functions of mulberry ethylene biosynthesis and signal pathway related genes in other aspects of the lifecycle remain unclear, especially in abiotic stress responses and tolerances. In this study, we investigated the physiological functions of a mulberry gene encoding EIN3-like proteins, *MnEIL3*, in salt and drought tolerances by analyzing its expression patterns and its heterologous overexpression in *Arabidopsis*. *MnEIL3*'s expression was significantly upregulated by salt and drought stresses, and its overexpression in *Arabidopsis* led to enhanced salt and drought stress tolerances and the upregulated expression of ethylene biosynthetic genes. Furthermore, MnEIL3 significantly enhanced the activities of *MnACO1* and *MnACS1* promoters. Thus, a working model for MnEIL3 in plant tolerance to abiotic stresses was suggested.

# MATERIALS AND METHODS

## Plant materials and growth conditions

*Arabidopsis thaliana* ecotype Columbia-0 and the *ein3-1 eil1-1* mutant were used as plant materials and were grown at 24 °C/22 °C under a 16-h light/8-h dark photoperiod.

A mulberry (*M. notabilis* Schneid) tree, which was used for genome sequencing, is an isolated wild mulberry species with a chromosome number of 14. The seedlings of *M. notabilis* were used in this study and grown in a PQX-type plant incubator with artificial intelligence capability (Ningbo Southeast Instrument Corporation, China) under a 16-h light/8-h dark photoperiod at 26 °C/22 °C (day/night). For stress treatments, the one-month-old seedlings were subjected to salt [0.6% (m/v) NaCl] and drought [20% (m/v) PEG6000]. The roots and shoot of the treated seedlings were sampled at 0, 1, 3, 6, 12, and 24 h post-treatment. The 14-day-old seedlings were treated independently with 200 mM NaCl and 200 mM mannitol, and the treated seedlings were sampled at 0, 1, 3, 6, and 12 h post-treatment. The harvested materials were frozen immediately in liquid nitrogen for total RNA extraction.

## RNA extraction and quantitative real-time PCR (qRT-PCR)

Total RNA extraction, first-strand of cDNA synthesis, and qRT-PCR analysis were performed as described in our previous study (*Wei et al., 2014*). The *ACTIN3* and β-*actin2* genes were used as internal controls for mulberry and *Arabidopsis*, respectively, and the relative expression was defined as $2^{-[Ct(target\ gene)\ Ct(control\ gene)]}$. All qRT-PCRs were performed with three independent biological replicates. The primers used are specified in Table S2.

## Plasmid construction

The full-length coding sequence of *MnEIL1* (GenBank accession number: XM_010107825) and *MnEIL3* (XM_010093690) were cloned into the *Nco*I and *Bgl*II restriction sites of the pCAMBIA1302 expression vector under the control of the CaMV35S promoter, and *MnEIL2* (XM_010107826) was cloned into the *Bgl*II and *Spe*I restriction sites of the pCAMBIA1302 expression vector. Finally, the *CaMV35S::MnEIL1*, *CaMV35S::MnEIL2*, and *CaMV35S::MnEIL3* recombinant plasmids were generated.

The 5′ upstream regions of the *MnACS1* and *MnACS3* genes were independently inserted into the *Eco*RI and *Nco*I restriction sites of the pCAMBIA1301 expression vector, producing the *MnACS1pro::GUS* and *MnACS3pro::GUS* recombinant plasmids, respectively. The primers used are specified in Table S2.

## Plant transformation

The recombinant plant expression vectors were transformed into *Agrobacterium tumefaciens* strain GV3101. *MnEIL3, MnACS1pro::GUS* and *MnACS3pro::GUS* were eventually independently transformed into *Arabidopsis thaliana* (Columbia-0) using the floral dip method (*Clough & Bent, 1998*). The transgenic *Arabidopsis* lines were evaluated by GUS staining, genomic PCR, inverse PCR, and qRT-PCR analyses. The homozygous lines of the T3 generation were used for further research.

## Stress treatments of transgenic *Arabidopsis*

Wild type*, ein3-1 eil1-1*, and *MnEIL3* transgenic seeds were germinated on 1/2 Murashige and Skoog (MS) agar medium. The 7-day-old seedlings were transferred into pots containing the soil supplemented with normal nutrients were grown at 24 °C/22 °C under a 16-h light/8-h dark photoperiod. The 21- and 14-day-old seedlings were treated with salt [1.2% (m/v) NaCl] and drought (watering treatments withheld), respectively, and the proline, hydrogen peroxide ($H_2O_2$), and malondialdehyde (MDA) contents, were measured using their respective test kits (Jiancheng Bioengineering Institute, Nanjing, China) according to the manufacturer's instructions. Each treatment was replicated three times.

The seeds of the transgenic plants that contained *MnACS1pro::GUS* and *MnACS3pro:: GUS* were germinated on 1/2 MS agar medium. The 10-day-old seedlings were exposed to salt (200 mM NaCl) and drought (200 mM mannitol) treatments. The samples were subjected to GUS staining after 0, 1, 3, 6, and 12 h of exposure.

## Transient expression assays and GUS activity detection

*CaMV35S::MnEIL1*, *CaMV35S::MnEIL2*, and *CaMV35S::MnEIL3* recombinant plasmids were used as effector plasmids, and *MnACO1pro::GUS*, *MnACO2pro::GUS*, *MnACS1pro::GUS*, and *MnACS3pro::GUS* recombinant plasmids were used as reporter plasmids. Of these vectors, the *MnACO1pro::GUS* and *MnACO2pro::GUS* recombinant plasmids have been reported in a previous study (*Yu et al., 2017*). The reporter and effector plasmids were transformed into *Agrobacterium tumefaciens* strain GV3101. The bacteria were mixed and co-injected into the strawberry fruit as described in a previous study (*Spolaore, Trainotti & Casadoro, 2001*). The injected tissues were sampled and used for the GUS staining analysis. Meanwhile, the GUS activity of the injected tissues was detected by 4-nitrophenyl-β-D-glucopyranoside methods (*Jefferson, 1987*).

## Statistical analyses

The statistical analyses methods were as described in a previous study (*Liu et al., 2017a*). All data were conducted using SPSS statistical software 17.0 (SPSS Inc., Chicago, IL, USA) and Excel 2013 (Microsoft, Redmond, CA, USA). The results are presented as mean values ± SEs. The significant differences between samples were analyzed using a one-way ANOVA in SPSS Statistics 17.0. The analyses of significant differences ($P < 0.05$) were measured by Student's *t*-test analysis.

# RESULTS

## Expression patterns of *MnEIL* genes under salt and drought stresses

The expression levels of *MnEIL* genes under NaCl and PEG treatments were assessed by qRT-PCR. Under salt stress, the transcript abundance of *MnEIL1* was significantly upregulated and downregulated in roots and shoots, respectively, 3 h after the NaCl treatment, but the expression levels after 24 h were not different than those at 0 h (Figs. 1A and 1B). The expression of *MnEIL2* in roots was significantly upregulated after 1 h of NaCl treatment. *MnEIL2*'s expression in shoots was downregulated at 1, 6, and 12 h, but its expression was upregulated at 24 h after the NaCl treatment (Figs. 1A and 1B). The transcript abundances of *MnEIL3* in roots and shoot were significantly upregulated 3 h after the NaCl treatment, but the expression of *MnEIL3* in shoots after 24 h showed no difference with that at 0 h (Figs. 1A and 1B). Under drought stress conditions, *MnEIL1*'s expression levels in roots and shoot were downregulated by the NaCl treatment, although *MnEIL1*'s expression in roots was upregulated at 3 h (Figs. 1C and 1D). The transcript abundance of *MnEIL2* was significantly upregulated and downregulated after 3 h of NaCl treatment in roots and shoot, respectively (Figs. 1C and 1D). The expression of *MnEIL3* in roots was significantly upregulated after a PEG treatment, although its expression showed no response to PEG at 12 h. *MnEIL3*'s expression in shoots was significantly upregulated 6 h after the PEG treatment and exhibited a strong expression peak at 24 h (Figs. 1C and 1D).

## The overexpression of *MnEIL3* in *Arabidopsis* enhances salt and drought tolerances

The *MnEIL3* gene was selected for further investigation base on its responses to salt and drought stresses. The full-length sequence of the *MnEIL3* gene was inserted into the

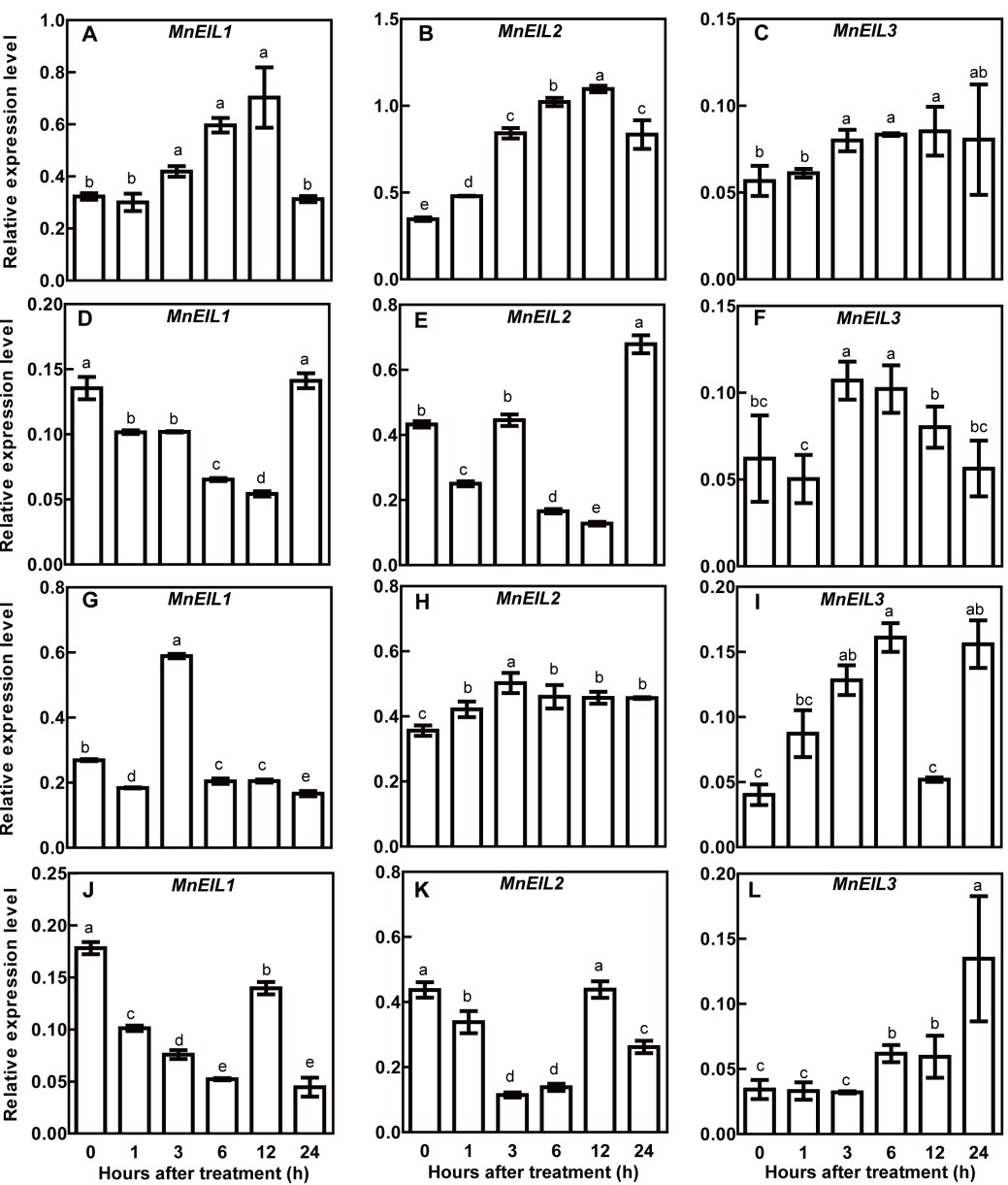

**Figure 1 Expression profiles of *MnEIL* genes in response to salt and drought stresses.** Mulberry seedlings were subjected to salt [0.6 % (m/v) NaCl] and drought [20% (m/v) PEG6000]. (A–C) *MnEIL* gene expression levels in roots under salt stress. (D–F) *MnEIL* gene expression levels in shoots under salt stress. (G–I) *MnEIL* gene expression levels in roots under drought stress. (J–L) *MnEIL* gene expression levels in shoots under drought stress. Data are means ± SEs ($n = 3$). Means within a column with different letters are significantly different ($P < 0.05$). Means within a column with the same letters are not significantly different ($P > 0.05$).

pCAMBIA1302 vector under the control of the *CaMV35S* promoter and transformed into wild type *Arabidopsis* plants. Transgenic lines were obtained using hygromycin resistance and confirmed by genomic PCR and qRT-PCR analyses (Figs. S1A and S1B). In addition, the insertion site of the transgene construct was determined using inverse PCR. The *CaMV35S::MnEIL3* recombinant plasmid was inserted into chromosome 2 of the *Arabidopsis* genome (Fig. S1C).

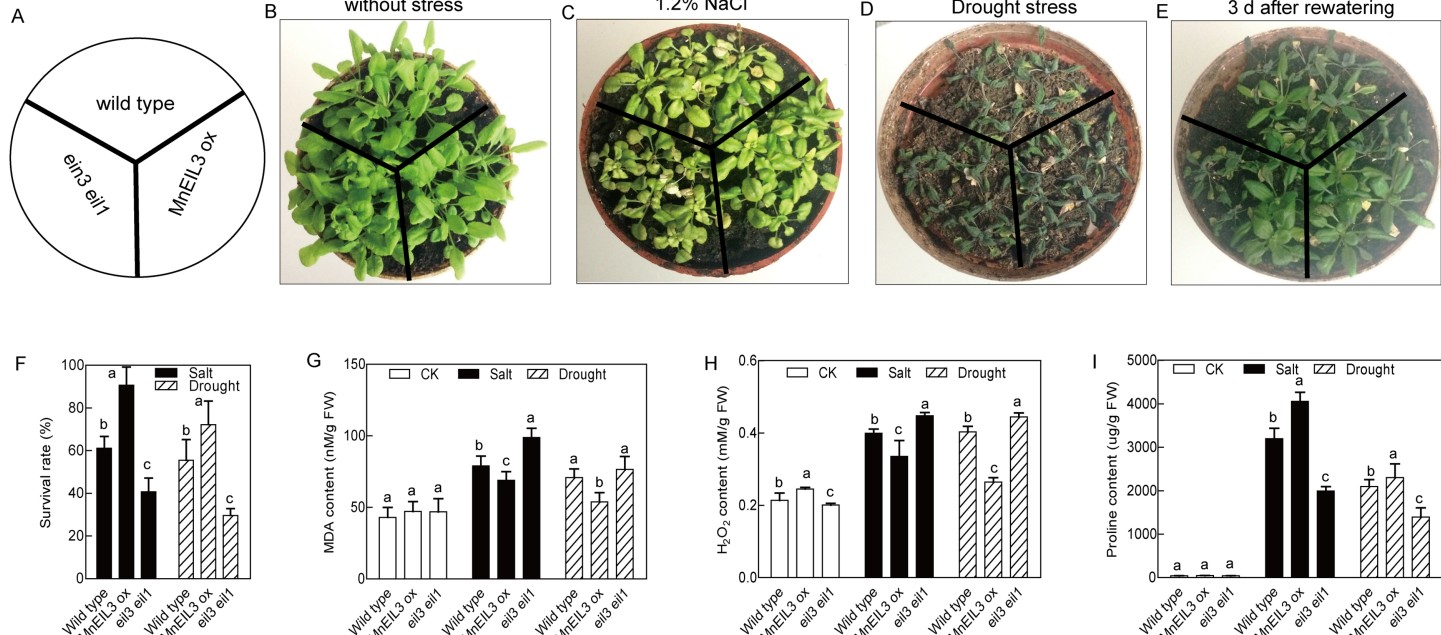

**Figure 2** **Stress tolerance analyses of wild type, *MnEIL3ox*, and *ein3-1 eil1-1 Arabidopsis*.** (A–E) The growth of wild type, *MnEIL3ox*, and *ein3-1 eil1-1 Arabidopsis* plants under normal, salt and drought conditions. (F–I) The survival rates (F), MDA content (G), $H_2O_2$ content (H), and proline content (I) of wild type, *MnEIL3ox*, and *ein3-1 eil1-1 Arabidopsi* plants under normal and stress conditions. Data are means ± SEs (*n* = 3). Means within a column with different letters are significantly different (*P* < 0.05). Means within a column with the same letters are not significantly different (*P* > 0.05).

To evaluate the tolerance of *MnEIL3*-overexpressing (*MnEIL3ox*) plants against salt stress, the 21-day-old wild type, *MnEIL3ox*, and *ein3-1 eil1-1 Arabidopsis* seedlings were treated with 1.2% (m/v) NaCl. After treatment, *MnEIL3ox* plants showed relatively greater growth rates than wild type and *ein3-1 eil1-1 Arabidopsis*. Additionally, the *ein3-1 eil1-1* seedlings showed a decreased salt tolerance compared with wild type plants (Figs. 2A and 2B). To characterize the performance of *MnEIL3ox* plants under drought stress, the 14-day-old seedlings of wild type, *MnEIL3ox*, and *ein3-1 eil1-1 Arabidopsis* plants were treated with drought stress. The growth of wild type, *MnEIL3ox*, and *ein3-1 eil1-1* plants showed no difference under drought stress conditions. However, *MnEIL3ox* showed a greater capability to survive than wild type and *ein3-1 eil1-1* plants when the treated plants were re-watered (Figs. 2A and 2B).

To understand the mechanism behind the enhanced sensitivity to drought and salt stresses caused by *MnEIL3*'s overexpression, the accumulated levels of $H_2O_2$, MDA, and proline were analyzed. The MDA and $H_2O_2$ contents in *MnEIL3ox* and *ein3-1 eil1-1* plants were lower and higher than in wild type *Arabidopsis*, respectively (Figs. 2C and 2D). The proline contents in *MnEIL3ox* and *ein3-1 eil1-1* plants were higher and lower than in wild type *Arabidopsis*, respectively (Figs. 2C and 2D). Thus, *MnEIL3* may negatively regulate drought and salt stress tolerances.

**The enhanced expressions of *ACS* and *ACO* genes in *MnEIL3ox* plants**

In this study, the expression levels of ACS- and ACO-encoding genes were detected in *MnEIL3ox* plants. The transcript abundances of *AtACS4/6/8/10/12* genes in *MnEIL3ox* plants were higher than in wild type *Arabidopsis*, while the expression levels of

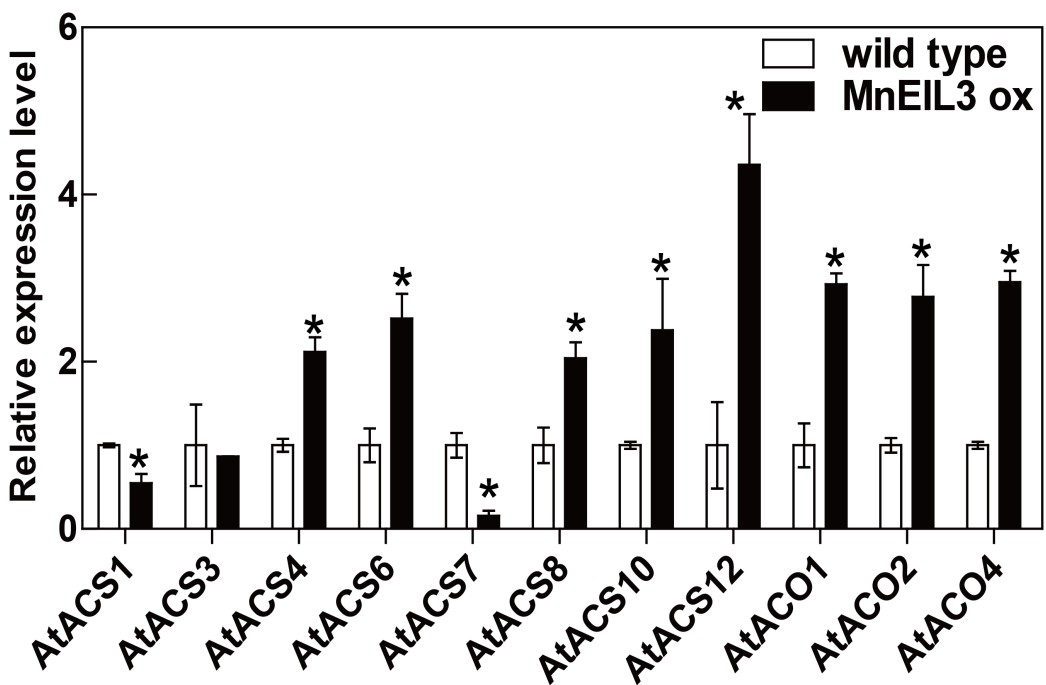

**Figure 3 qRT-PCR analysis of changes in *AtACS* and *AtACO* genes in *MnEIL3ox* plants.** Data are means ± SEs (*n* = 3). Significant differences (*P* < 0.05) are marked with asterisks. The gene expression in wild type was set as 1.      

*AtACS1/7* genes showed were lower in *MnEIL3ox* plants. Moreover, there was no difference in the expression of *AtACS3* between *MnEIL3ox* and wild type plants (Fig. 3). All of the detected *AtACO* genes, *AtACO1/2/4*, showed higher expression levels in *MnEIL3ox* plants than in wild type *Arabidopsis* (Fig. 3).

## Mulberry EIL proteins modulate *MnACO1*, *MnACO2*, *MnACS1*, and *MnACS3* promoter activities

To explore the correlations between the expression of *MnEIL3* and ethylene biosynthetic genes, *MnACO1*, *MnACO2*, *MnACS1*, and *MnACS3* were selected for promoter isolation. All these genes have been determined as the key genes involved in ethylene biosynthesis in mulberry (*Liu et al., 2014, 2015*). The gene' promoters were downloaded from the *Morus* genome database (http://morus.swu.edu.cn/morusdb/) and isolated from mulberry (*M. notabilis*). By searching for cis-acting regulatory elements, PERE- and ECBS-binding sites were identified in the promoters of *MnACO1*, *MnACO2*, *MnACS1*, and *MnACS3* (Table S1). In vivo interactions between *MnEIL3* and these promoters were estimated by transient analyses in strawberry fruit. *MnEIL3* significantly enhanced the activities of *MnACO1* and *MnACS1* promoters, while no significant effects on the activities of *MnACO2* and *MnACS3* promoters were found (Fig. 4). Thus, *EIL* proteins may act as the transcriptional activators of ethylene biosynthetic genes. We also detected correlations between the other two *MnEIL* genes, *MnEIL1/2*, and ethylene biosynthetic genes. Thus, *MnEIL1* acted as the activator of *MnACO2* and *MnACS3* promoters, while *MnEIL2* regulated the activities of the *MnACO1* and *MnACS3* promoters (Fig. 4).

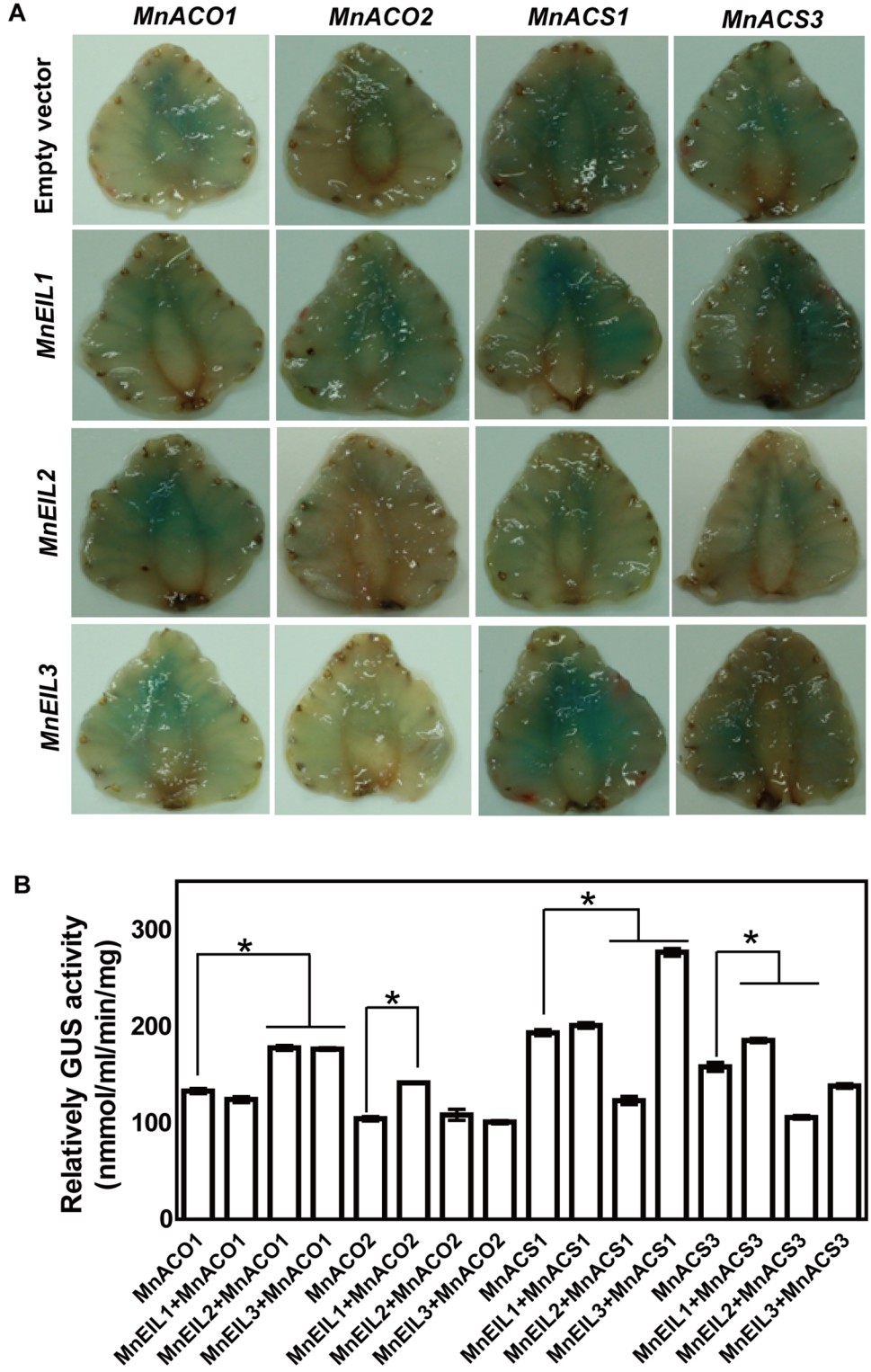

**Figure 4 In vivo interactions of MnEIL3 with ethylene biosythetic genes promoters.** (A) GUS staining of the injected strawberry fruit. (B) The detection of GUS activities. Data are means ± SEs ($n = 3$). Significant differences ($P < 0.05$) are marked with asterisks.

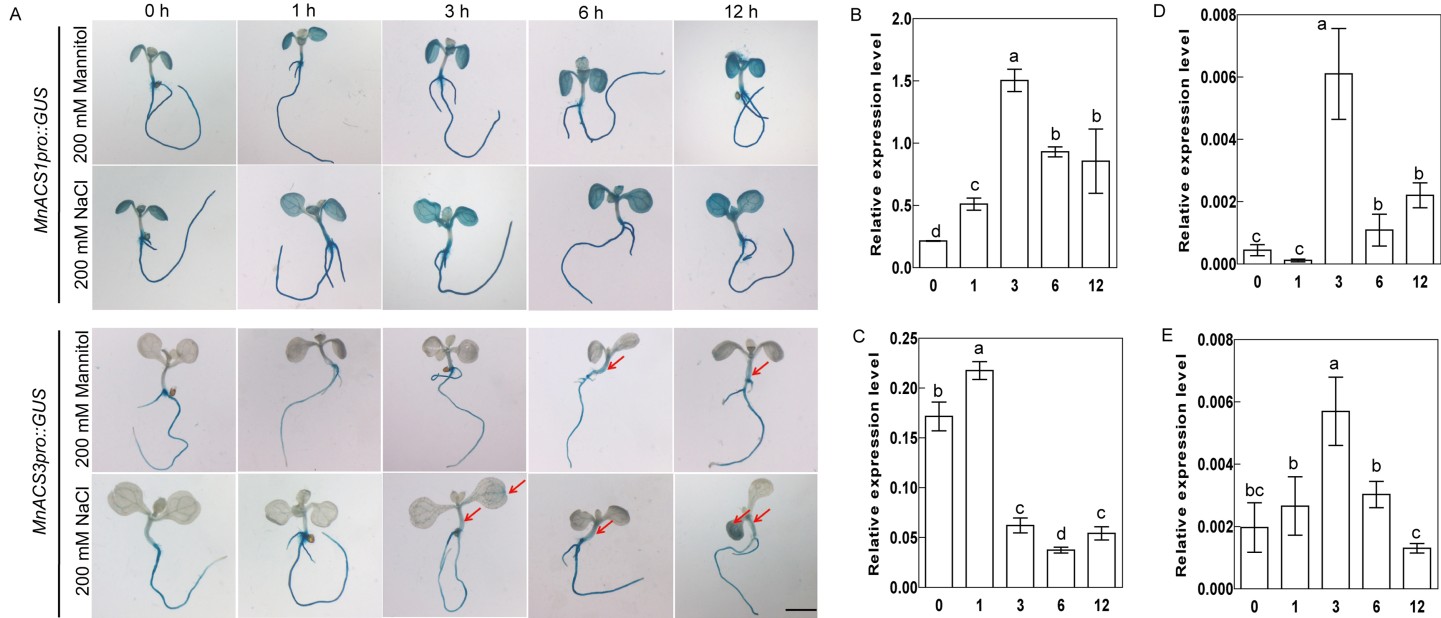

**Figure 5 Analyses of promoter activities and expression levels of *MnACS1* and *MaACS3* genes in response to salt and drought treatments.** (A) Histochemical GUS staining of the MnACS1pro::GUS and MnACS3pro::GUS transgenic *Arabidopsis* under salt [200 mM NaCl] and drought (200 mM mannitol) treatments. Scale bar, two mm. (B) The expression level of *MnACS1* in response to salt treatment. (C) The expression level of *MnACS1* in response to drought treatment. (D) The expression level of *MnACS3* in response to salt treatment. (E) The expression level of *MnACS3* in response to drought treatment. Data are means ± SEs (*n* = 3). Means within a column with different letters are significantly different (*P* < 0.05). Means within a column with the same letters are not significantly different (*P* > 0.05).

## The activities of *MnACS1* and *MnACS3* promoters were regulated by salt and drought stresses

To examine the responsiveness of the *MnACS1* and *MnACS3* genes under salt and drought stresses, transgenic *Arabidopsis* were generated by introducing *MnACS1pro::GUS-* and *MnACS3pro::GUS*-fused genes, and then the 10-day-old seedlings were exposed to stresses (Fig. 5A). The *MnACS1* promoter in leaves responded to NaCl and mannitol treatments (Fig. 5A). The GUS reporter in the *MnACS3pro::GUS* transgenic *Arabidopsis* was mainly expressed in roots, while little GUS accumulation levels in stems and leaves were detected. Under salt stress, the GUS accumulation levels in the stems of *MnACS3pro:: GUS* transgenic *Arabidopsis* were enhanced after 6 h of NaCl treatment, while the GUS levels in stems and leaves were enhanced after 3 h of mannitol treatment (Fig. 5A).

## The expression levels of *MnACS1* and *MnACS3* are regulated by salt and drought stresses

The 14-day-old seedlings were treated with NaCl and mannitol, and then used to detect the expression levels of *MnACS1* and *MnACS3*. *MnACS1*'s expression level was significantly upregulated after 1 h of NaCl treatment, and its expression was transiently upregulated after 1 h of mannitol treatment and then was downregulated (Figs. 5B and 5C). The transcript abundance of *MnACS3* was significantly upregulated after 3 h of NaCl treatment, and its expression was upregulated after 3 h of mannitol treatment, but it showed no response at any other time point (Figs. 5D and 5E).

## DISCUSSION

EIN3/EILs proteins are positive factors in ethylene signal transduction. In model plants, EIN3/EILs are involved in many aspects of the life cycle, including seed germination, soil emergence, seedling development, leaf senescence, pigments biosynthesis, light perception, and abiotic stress responses (*An et al., 2018*; *Kim, Cho & Yoo, 2017*; *Yu et al., 2013*; *Yu et al., 2016*; *Zhong et al., 2014*). Additionally, the regulatory functions of EIN3/EILs in response to abiotic stresses have also attracted considerable attention (*Cui et al., 2015*; *Kong et al., 2018*; *Peng et al., 2014*; *Shi et al., 2012*; *Zhang et al., 2011*). However, there are no reports on the functions of EIN3/EILs in the abiotic stress tolerance of woody plants, including mulberry. In the present study, the expression levels of mulberry *MnEIL* genes under salt and drought stresses were revealed, and they showed different patterns. Among these genes, the expression of *MnEIL3* was significantly upregulated by salt and drought stresses in roots and shoots (Fig. 1), which is similar to the expression patterns of *Arabidopsis AtEIN3* and *AtEIL1* genes (Fig. S2). The full-length coding sequence of *MnEIL3* was transformed into *Arabidopsis* for stress tolerance analysis. The expression levels of *MnEIL3* were significantly upregulated in roots and shoots by NaCl treatments (Fig. 1), and the overexpression of this gene in *Arabidopsis* enhanced salt stress tolerance (Fig. 2). *MnEIL3*'s overexpression decreased the MDA and $H_2O_2$ contents and enhanced the proline content under salt stress. Thus, *MnEIL3* positively regulated plant salt tolerance, which was similar to the results described by *Peng et al. (2014)*. In this study, *ein3-1 eil1-1* plants showed decreased tolerances to drought stress compared with wild type plants (Fig. 2), and mutation of EIN3/EILs decreased the MDA and $H_2O_2$ contents and enhanced the proline content under drought stress. This was similar to the results of a previous study which reported that the *ein3-1* mutant exhibited a decreased tolerance to drought stress stimulated by PEG6000 (*Cui et al., 2015*). In addition, *MnEIL3ox* plants showed a greater ability to survive drought stress than wild type and *ein3-1 eil1-1* plants. Thus, MnEIL3 may play a positive role in abiotic stress tolerances, and it indicates that the functions of EIL3/EILs in response to abiotic stresses are relatively conserved in plants.

The overexpression of kiwifruit (*Actinidia deliciosa*) EIN3-like TFs, *AdEIL2* and *AdEIL3*, increased ethylene production by upregulating the expression of *ACS* and *ACO* genes in transgenic *Arabidopsis* (*Yin et al., 2010*). Based on the data reported by *Liu et al. (2017b)*, several genes that were involved in ethylene biosynthesis and signal transduction showed lower expression levels in *ein3-1 eil1-1* plants compared with wild type *Arabidopsis*. Here, the transcript abundances of *AtACS* and *AtACO* genes were mainly upregulated in *MnEIL3ox* plants. The analysis suggested the positive feedback regulation of EIN3/EILs in ethylene production.

When plants receive the ethylene signal, EIN3/EILs are activated, and then, they regulate the transcriptional expression of downstream responsive genes, including ERF, ACO, xyloglucan endo-transglycosylase, and cell wall-modifying genes (*Huang et al., 2010*; *Ireland et al., 2014*; *Solano et al., 1998*; *Yin et al., 2010*). The PERE and ECBS motifs in promoters have been identified as EIN3-interactive motifs. *MnACO1*, *MnACO2*, *MnACS1*, and *MnACS3* promoters contain PERE and ECBS motifs, which suggests that

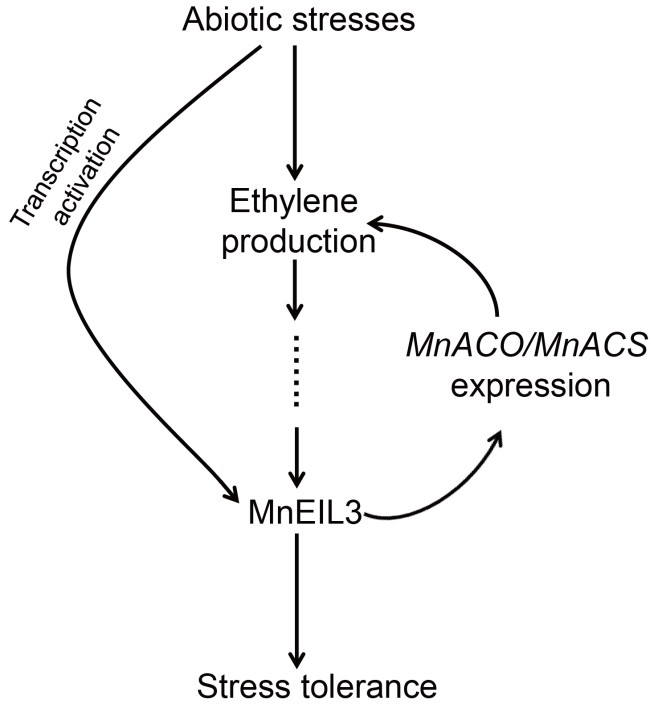

**Figure 6 A possible model of *MnEIL3* in responses to abiotic stresses.**

ethylene biosynthetic genes can be regulated by EIN3/EILs. In the present study, we found that MnEIL3 and two other MnEILs (MnEIL1 and MnEIL2) modulate the activities of *MnACO1* and *MnACO2* promoters as assessed by transient analysis in strawberry (*Fragaria × ananassa* Duch.) fruit. This result was similar to those reported in kiwifruit and melon (*Cucumis melo* L. cv. *Andes*) (*Huang et al., 2010*; *Yin et al., 2010*). *MnACS1* and *MnACS3* promoter' activities were regulated by MnEIL1/2/3 (Fig. 4). Thus, MnEIL proteins provide positive feedback regulation during ethylene production by directly regulating the transcription of ethylene biosynthetic genes. We also constructed *MnACS1pro::GUS* and *MnACS3pro::GUS* vectors and independently introduced them into *Arabidopsis*. The GUS activities in the stems and leaves of the transgenic *Arabidopsis* seedlings were enhanced by salt and drought stresses (Fig. 5A). Additionally, the expression levels of *MnACS1* and *MnACS3* significantly responded to salt and drought stresses (Figs. 5B–5E). The promoter' activities and gene expression levels of *MnACO1* and *MnACO2* were also enhanced by abiotic stresses (*Yu et al., 2017*).

On the basis of our results, we proposed a working model for the regulatory network of mulberry MnEIL3 in response to abiotic stresses (Fig. 6). When plants are exposed to abiotic stresses, the stress signals are perceived by plant cells, leading to the enhanced accumulation of ethylene. Then, the ethylene signal transduction pathway was induced and the nucleus-localized MnEIL3 proteins' accumulation and stability were enhanced. *MnEIL3*'s expression was also significantly induced by stresses. MnEIL3 eventually positively regulates abiotic stress tolerances by activating downstream stress-responsive genes. In addition, MnEIL3 binds to the target regions in the promoters of *ACO* and

*ACS* genes, and activates gene expression, which contributes to the accumulation of ethylene. MnEIL1 and MnEIL2 may function in other processes, such as fruit development and maturation, by modulating ethylene responses (*Liu et al., 2015*). However, more work is needed to investigate the roles of the ethylene–EIN3/EILs–*ACO*/*ACS* regulatory loop in abiotic stress tolerances.

## CONCLUSIONS

In summary, our results explored the functions of a gene encoding an EIN3-like protein from mulberry, *MnEIL3*. The expression level of *MnEIL3* significantly increased in response to salt and drought stresses in roots and shoot. Transgenic *Arabidopsis* overexpressing *MnEIL3* exhibited an enhanced tolerance to salt and drought stresses. The overexpression of *MnEIL3* significantly upregulated the expression levels of ethylene biosynthetic genes in *Arabidopsis*. Moreover, *MnEIL3* could enhance the activities of *MnACO1* and *MnACS1* promoters, which suggested an ethylene–EIN3/EILs–*ACO*/*ACS* regulatory loop in abiotic stress tolerance. This research provides insights into the functions of MnEIL3 in abiotic stress tolerance and their influence on the expression levels of ethylene biosynthetic genes.

## ACKNOWLEDGEMENTS

We thank Dr. Hongwei Guo (Institute of Plant and Food Science, Department of Biology, Southern University of Science and Technology, Shenzhen 518055, P.R. China) for providing the seeds of *ein3-1 eil1-1 Arabidopsis*.

### Funding

This work was supported by the China Agriculture Research System (No. CARS-18–ZJ0201), the Special Fund for Agro-scientific Research in the Public Interest of China (No. 201403064), the Fundamental Research Funds for the Central Universities (No. XDJK2018C008), and the National Natural Sciences Foundation of China (Grant No. 31801126). The funders had no role in study design, data collection and analysis, decision to publish, or preparation of the manuscript.

### Grant Disclosures

The following grant information was disclosed by the authors:
The China Agriculture Research System: CARS-18–ZJ0201.
The Special Fund for Agro-scientific Research in the Public Interest of China: 201403064.
The Fundamental Research Funds for the Central Universities: XDJK2018C008.
The National Natural Sciences Foundation of China: 31801126.

### Competing Interests

The authors declare that they have no competing interests.

## Author Contributions

- Changying Liu conceived and designed the experiments, performed the experiments, analyzed the data, prepared figures and/or tables, authored or reviewed drafts of the paper, approved the final draft.
- Jun Li performed the experiments.
- Panpan Zhu performed the experiments.
- Jian Yu performed the experiments.
- Jiamin Hou performed the experiments.
- Chuanhong Wang performed the experiments.
- Dingpei Long analyzed the data.
- Maode Yu conceived and designed the experiments.
- Aichun Zhao conceived and designed the experiments, contributed reagents/materials/analysis tools, authored or reviewed drafts of the paper, approved the final draft.

## Data Availability

The raw data are available in Data S1.

## Supplemental Information

Supplemental information for this article can be found online at http://dx.doi.org/10.7717/peerj.6391#supplemental-information.

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
