# Peer review of "Mulberry EIL3 confers salt and drought tolerances and modulates ethylene biosynthetic gene expression"

_PeerJ, doi:10.7717/peerj.6391_

## Round 0.1 · original submission · Major Revisions

Please address all the critical issues raised by three reviewers and revise your manuscript accordingly.

Reviewer 1 ·

Basic reporting

No comment.

Experimental design

No comment.

Validity of the findings

No comment.

Additional comments

Mulberry is an important economically plant which has a very wide distribution range in world, especially in China. In this manuscript, the authors investigate the function of EIN3/EIL, the key component involve in ethylene signal transduction, in response to abiotic stresses in non-model plant mulberry. Using quantitative real-time PCR analysis, the authors found that expression levels of MnEIL3 were significantly upregulated in roots and shoot in response to salt and drought stress. Meanwhile, overexpression of this gene in Arabidopsis enhanced plant’s tolerance to salt stresses and showed better ability to survive from drought stress after re-watering. Further, authors tested the influence of MnEIL3 on the transcription of four downstream ethylene biosynthetic genes and showed that MnEIL3 significantly enhanced the activities of two promotors. Based on their experimental results, a working model of mulberry MnEIL3 in response to abiotic stress was proposed.

This manuscript is well organized and well written. The authors familiar with the background and progress in the field. Characterization of the function of MnEIL3 in abiotic stresses tolerance and ethylene biosynthetic genes expression will provide new clues for the community. I am in favor of this study being published in PeerJ after the “Discussion” be revised by adding the comparison of the function of EIL3/EIL in response to abiotic stresses in model plants and non-model plants. Is there any difference? Another suggestion is revision of the title. The title should be concise, provide an overall view of the paper’s significance.

Typo mistake:
Line240 : “on” significant differences should be “no” significant difference.

Reviewer 2 ·

Basic reporting

This manuscript was clearly written and easy to follow. The Introduction section provided sufficient background information of the research field with proper literature references. Minor wording issues: line 42, “ethylene and resulting in”, should be “ethylene, resulting in”; line 57, “stabilized” should be “stabilizes”; line 236, “enhanced the tolerance to salt stresses” should be “enhanced the tolerance to salt stresses (Fig. 2)”; line 240-241, should correct to “no significant differences appeared between the growth of”; line 247-248, “the data has been reported” should be “the data reported”; line 251, should be “The analysis suggested”; line 262, should be “results reported in kiwifruit”; line 264, “providing” should be “provide”; line 268, “(Fig. S2)” should be “(Fig. S3)”; line 284, “overexpressed” should be “overexpressing”. It would be better if the authors can mention in Introduction that ein3-1 and eil1-1 are ethylene-insensitive mutants. Moreover, to remain consistent, “ein3 eil1 Arabidopsis” should be referred to as “ein3-1 eil1-1 Arabidopsis”. As for figure legends, the authors should add what significance levels letters “a”, “b” and “c” refer to in Fig.1.

Experimental design

Sufficient details were provided in the Materials and Methods section as well as supplementary materials to allow for replication. It would be nice if the authors can explain the rationale as for why MnEIL3 was chosen as the focus of the study, not MnEIL1 or MnEIL2. The fact that MnEIL1/2 were able to modulate the activities of MnACO/MnACS promoters as demonstrated in the strawberry transient analysis is a sign that MnEIL1/2 could also play important roles in ethylene signal transduction. If this is the case, it will be more informative if the authors can evaluate the expression profiles of MnEIL1/2 in response to salt/drought stresses.

Validity of the findings

In general, the data presented in this manuscript are robust with experimental replicates and supported the points the authors tried to make. One minor issue with Fig.3, the error bar was not clear for AtACS3. Were the numbers too tight to produce error bars or not enough replicates existed? In the Results section, although the authors stated that “the activities of MnACS1 and MnACS3 promoters were regulated by salt and drought stresses”, it was not clear from Fig. S3 how the MnACS1 promoter showed responses to abiotic stresses. Therefore it will be helpful if the authors can elaborate in the Results section how MnACS1 was regulated by abiotic stresses.

Additional comments

1. The authors showed enhanced expression of AtACS/ACO genes in MnEIL3ox Arabidopsis as well as increased activities of MnACS promoters in Arabidopsis induced by salt/drought stresses. If possible, it will be better to measure the MnACS/ACO expression levels in Mulberry and see how they change upon salt/drought stresses.
2. In Fig.1, the MnEIL3 expression showed no significant differences at selected time points compared to no treatment. It will be nice if the authors can provide some hypothesis/speculations in the Discussion section as for why this might be the case. Or were similar time-dependent profiles previously observed in model plants for EIN3/EILs genes?

Reviewer 3 ·

Basic reporting

This manuscript of Liu et al. describes the effects of the MnEIL3 gene on salt tolerance and drought stresses in Morus notabilis. They analyzed the role of MnEIL3 by heterologous overexpression in Arabidopsis by measuring resistance to salt and drought stress and induction of ethylene biosynthesis genes.The data provided in the manuscript are interesting, but unfortunately, the results presented are not sufficiently precise and rigorous to be accepted as is.

A native English speaker must revise the English language of this article.

The introduction of the manuscript is clear and the working hypothesis and objectives clearly exposed. However, there is an overall bias in the paper with no mention of occidental researches that are at the origin of the major discoveries on the ethylene biosynthesis and signalling pathways. Despite the fact that this work is impressive, the presentation of the work lacks precision and consistency in its formal construction.

There is confusion for an uninitiated person between Morus alba L. (line 86) and Morus notabilis Schneid (line 106). Are alba and notabilis the same specie? Are they different species? Why are genes noted with MnEIL3 instead of MaEIL3? Can the authors present the genus Morus more clearly for an uninitiated reader? Which genus Morus model species have been already sequenced?

Throughout the article several acronyms are not defined: TFs, SIEDS, AP2, ESE1, CBFs, ARR, XTH33, LSUI, PERE, ECBS, ...
Finally, some major results are presented in the additional data. However, they are cited in the result section as major results. Why are not they integrated directly into the result section? There is therefore a major problem of coherence in the construction of the manuscript( ex: line 195-222).
There is an absence of homogeneity in the references: ex: line 430-435

Experimental design

This paper answers to the aim and scope of the journal.

However, the material and methods section suffers from a lack of precision. For example, there is no mention of the number of technical and biological replicates performed during PCRs and housekeeping genes used as control. How the calculations are made? Similarly, in the section statistical analysis, the nonparametric or parametric tests used are not indicated: T-test, Kruskal-Wallis, Mood's median test, ANOVA, ... nor the number of repetitions used.

I have a personal problem of understanding this study in relation to the induction of salt and water stresses that are never done on plants of the same age and in the same way. In my opinion the results are therefore difficult to compare. For example, in material and method section, see the different ways to get water stress: line 109 line 112 line 137 line 141. How the authors can justify these differences in the plant treatment?

Furthermore, Figure 2 that is supposed to show the effects of salt and water stresses is very poor scientifically. There is no serious quantified evaluation. Why did the authors not quantify the results of drought stress resistance by measuring the changes of proline and ABA concentrations, water loss of detached leaves, stomatal conductances, transpiration rate,...(see for ex: Yu, H., et al. Plant Cell 2008; 20:1134-1151)?

Validity of the findings

As mentioned earlier, the work lacks quantified results of the effects of water stress and drought on the transformed plants. This seriously affects the strength of the conclusions and the hypothetic model based solely on the expression data.
It is imperative that the PCR data be better explained (number of biological and / or technical repeats, control housekeeping genes, calculations...) and the statistics used indicated.

Additional comments

In conclusion, the publication cannot be reasonably accepted in this state without being rewritten and without additional data to better quantify at physiological level the effects of salt and water stresses in arabidopsis and Morus notabilis.

---

## Round 0.2 · accepted · Accept

Thank you for addressing all critical issues pointed by the reviewers and for the careful revision of the manuscript.

# Reviewer 1 ·

Basic reporting

No comment.

Experimental design

No comment.

Validity of the findings

No comment.

Additional comments

The authors revised the title and text according to reviewers’ comments. Their efforts to provide more evidence to support the findings are appreciated. At the same time, detailed descriptions of methods and figure legends make the manuscript clear. The present result is recommended to publish.

Reviewer 2 ·

Basic reporting

The authors revised their wording according to reviewers' comments. Their efforts to proofread the language by an editor should be appreciated.

Experimental design

The authors addressed my comments really well by elaborating on their choice of using MnEIL3 and by incorporating the expression profiles of MnEIL1/2 in Figure 1.

Validity of the findings

The authors addressed my comments by providing further descriptions of GUS activities (MnACS1) in transgenic Arabidopsis seedlings.

Additional comments

I really appreciate the authors' efforts to add MnACS/ACO expression profiles to Figure 5, which helped support their conclusions. Although it will be better, as I suggested in my original comments, to provide MnACS/ACO expression profiles and how they respond to salt/drought stresses in Mulberry instead of the model plant Arabidopsis. I understand that this data could be challenging to obtain, but would make the conclusions much stronger if available.
It's very nice that the authors provided the expression profiles of AtEIN3 and AtEIL1, which help address my comments.